# Functional Benefit of Smoking Cessation and Triple Inhaler in Combustible Cigarette Smokers with Severe COPD: A Retrospective Study

**DOI:** 10.3390/jcm12010234

**Published:** 2022-12-28

**Authors:** Aldo Pezzuto, Giuseppe Tonini, Massimo Ciccozzi, Pierfilippo Crucitti, Michela D’Ascanio, Fiammetta Cosci, Antonella Tammaro, Antonella Di Sotto, Teresa Palermo, Elisabetta Carico, Alberto Ricci

**Affiliations:** 1Cardiovascular-Respiratory Sciences Department, Sant’Andrea Hospital-Sapienza University, Via di Grottarossa, 1035/39, 00189 Rome, Italy; 2Department of Oncology Campus, Bio-Medico University, 00197 Rome, Italy; 3Department of Epidemiology Campus, Bio-Medico University, 00197 Rome, Italy; 4Department of General Surgery Campus, Bio-Medico University, 00197 Rome, Italy; 5Department of Health Sciences, University of Florence, 50121 Florence, Italy; 6Department of Neurosciences, Mental Health and Sensory Organs [NESMOS], Sapienza University, 00189 Rome, Italy; 7Department of Physiology and Pharmacology, “Vittorio Erspamer”—Sapienza University, 00128 Rome, Italy; 8Clinical and Molecular Medicine Department, Sapienza University, 00189 Rome, Italy

**Keywords:** smoking cessation, severe COPD, triple therapy

## Abstract

Introduction: Chronic obstructive pulmonary disease (COPD) is the third cause of mortality and it is smoking-related. It is characterized by a non-reversible airflow limitation and a progressive worsening of the respiratory function. Objective: The aim of this study is to point out the benefit of smoking cessation combined with a single inhaler triple therapy in terms of clinical and functional outcome in this setting. Methods: A retrospective analysis was performed in patients affected by severe COPD and at least one exacerbation a year, who underwent a smoking cessation program. All patients underwent a 6 min walking test, body plethysmography, and an exhaled test for carbon monoxide. The modified medical research council test (mMRC) test, the Fagestrom nicotine dependency test (FTND) and the COPD assessment test (CAT) questionnaire were also administered. All patients were checked at the baseline and in the six-month follow-up after the start of the treatment. Results: Smoking cessation was achieved by 51% of patients within a month and it was confirmed by eCO measure (<7 ppm). Patients who quit smoking reported better results after six months compared with patients who did not. The increase in FEV1 within the group of quitters was 90 mL (*p* < 0.05) and the walking test improved by 90 m (*p* < 0.01); eCO decreased by 15 ppm (*p* < 0.01) while FVC increased by 70 mL (*p* < 0.05). No significant changes were recorded within the group of sustainers. The difference in functional changes between groups was significant with regard to FEV1, cCO, and WT. Conclusions: Smoking cessation enhances the efficacy of single inhaler triple therapy, improving clinical and functional variables after six months from the start.

## 1. Introduction

Chronic obstructive pulmonary disease (COPD) is a disease characterized by airflow limitation with a progressive decline in lung function over time, especially in smokers, and it is often complicated by exacerbations which are defined as a worsening of patients’ respiratory symptoms leading to the need for oral corticosteroids, antibiotics, or hospitalization [1,2,3].

Smoking cessation is the main approach that can change the course of the disease, reducing the annual rate of FEV1 decline, and it is effective in the short term in improving symptoms and respiratory function [4].

Patients with severe COPD combined with a history of exacerbations may require a LABA/LAMA double combination which is more effective than ICS/LABA at reducing the annual exacerbation rate [5].

FEV1 changes are the primary functional marker associated with the severity of the disease [6].

It is known that the number and the severity of exacerbations per year could affect mortality [7].

The aim of the current study is to point out the effects of smoking cessation along with a single inhaler triple therapy on respiratory functional parameters after six months in severe COPD patients, comparing quitters with patients who continue to smoke (sustainers).

## 2. Methods

### 2.1. Study Population

A retrospective analysis was performed on a population of 45 smokers affected by severe COPD. Severe COPD was determined based on a FEV/FVC ratio of < 70% and post-bronchodilator FEV1 of < 50% predicted according to GOLD guidelines [8].

Two subgroups were compared, one of quitters [23 patients] and one of sustainers [22 patients], who only reduced cigarette consumption without quitting. The smoking cessation outcome was assessed via exhaled CO detection.

Exclusion criteria were hospitalization within four weeks prior to the first evaluation, severe comorbidities, and a history of asthma.

The study was approved by the Sapienza-S. Andrea Ethics Committee whose registration number is 773. An informed consent was obtained from each patient.

### 2.2. Study Design

The functional analysis was performed at the baseline and at six months after the start of the smoking cessation treatment.

All patients underwent treatment with beclometasone-formoterol-glycopirronium (BDP/FF/G)87/5/9 µg single inhaler with MDI device at a dosage of two inhalations twice a day along with smoking cessation front-line treatment.

### 2.3. Assessment and Smoking Cessation

Smoking cessation consisted of 20 min counselling plus varenicline administration, which is a α4β2 nicotinic-acetylcholine receptor partial agonist. Counselling was based on the five “A”s: Ask about smoking habits; Advise patient to quit smoking; Assess offering assistance; Assist with a quit plan; Arrange scheduling follow-up. The schedule dosage of varenicline was 0.5 mg per day for three days, then 0.5 mg twice a day for four days, increasing the dosage up to 1 mg twice a day. A patient quitter was defined as a subject who reported an eCO level equal to or lower than 7 ppm. Some tests were administered: the Fagestrom test for nicotine dependence (FTND) (range 0–2 no dependence, 3–4 low, 5–7 moderate, 8–10 high dependence), the CAT questionnaire (range 0–40) [9], and an mMRC test (range 0–4) [10]. Concerning spirometry, post-bronchodilation forced expiratory one second volume (FEV1), forced vital capacity (FVC) and peripheral flow expiratory forced (FEF) 25/75% were recorded. A 6 min walking test (WT) was also performed, during which the oxygen saturation and the distance covered were recorded by PalmSAT 2500 oxymeter (Nonin Med Inc., Plymouth, MN, USA) [11]. The exhaled CO measurement was detected by Smokerlyzer with an electrochemichal sensor (Bedfont, USA) [12]. The normal value was set below 7 ppm.

The spirometry was performed by body plethysmography following the ATS and ERS Task Force 2005 guidelines (Jaeger system masterscreen, Germany). Bronchodilation was carried out by having the smokers inhale 400 g of salbutamol. The techniques followed the American Thoracic Society and European Respiratory Society task force guidelines [13].

## 3. Statistical Analysis

All data are expressed as median and interquartile ranges, or the mean and SD as appropriate.

Comparison between groups at the baseline was determined by the Mann–Whitney U test for continuous variables and by a contingency table for categorical ones. The Wilcoxon rank sign test was applied for comparison within the groups.

The comparison of functional changes among the groups was detected by the Mann–Whitney U test.

Statistical significance was assumed when null hypothesis could be rejected at *p* < 0.05. Statistical analysis was carried out by SPSS 24.0 for Windows (Chicago, IL, USA).

## 4. Results

Table 1 shows the baseline demographic and functional data.

Participants had an average age of 55.5 years with a prevalence of males, a mean pack-years of 25. The average BMI was 30.7 and exhaled CO was 23 ppm. The FTND basal mean value was 5.0 and FEV1 1.12 l. FVC was 1.93 and 70% of predicted.

In Table 2, the changes in values from baseline to six months after the start of the therapy are reported.

No significant differences were recorded in terms of age and BMI.

The variations within the group of quitters were significant: FEV1 increased by 90 mL and FVC increased by 70 mL (*p* < 0.05), the walking test and eCO also improved significantly by 90 m and 15 ppm, respectively (*p* < 0.01). The CAT value was also improved though not significantly. A FEF of 25/75 did not show any substantial variation.

The FEV1 percentage of the predicted value increased by 7%, whereas the FVC increased by 6% and this increase was significant (*p* < 0.05). No significant variations were observed in the group of sustainers with a lower improvement in FEV1 and in the walking test.

In Table 3, the changes in FEV1, eCO, and WT are analyzed, comparing quitters with sustainers; the differences were significant (*p* < 0.005, *p* < 0.003, *p* < 0.001).

## 5. Discussion

This study shows the effectiveness of single inhaler triple therapy combined with smoking cessation treatment in smokers.

As expected, patients who quit smoking achieved better clinical and functional results than those who reduced but did not quit.

In the six-month follow-up, important symptoms such as dyspnea and others included in the CAT test improved significantly along with functional indices.

Relieving respiratory symptoms is the cornerstone to improving quality of life. Smoking cessation is strongly recommended as well as bronchodilation treatment.

According to the literature, extrafine BDP/FF/G showed its efficacy in improving quality of life by prolonging the time to the first important deterioration compared with a single and double bronchodilation and ICS/LABA [14].

Triple single inhalers showed a reduction in moderate-severe exacerbations rate compared with dual bronchodilation [15].

The extrafine single inhaler triple therapy showed its benefits in terms of exacerbation rate with an improvement in functional parameters [16,17,18,19]. Cigarette smoke is made up of more than 4000 recognized substances, among which are gases and solid ones; all of these are potentially dangerous causing bronchial inflammation and carcinogenesis. Smoking cessation is the most important effective intervention to prevent both chronic bronchial obstruction and lung nodules evolution [20,21].

The natural history of COPD is characterized by flow limitation with a progressive functional decline; it is also worsened by exacerbations [22].

Lung function decline is closely linked with age and smoking habits which bring on worsening symptoms. In our study, a significant variation in all parameters was observed only in patients who quit smoking, which shows that triple therapy is more effective in this type of subject.

This study has limitations due to the retrospective nature on a small sample of patients.

Varenicline, a α4-β2 acetyl-choline partial agonist, confirmed its validity in terms of smoking cessation and indirectly by favoring a reduction in bronchial inflammation [23,24]. The latter showed its efficacy as front-line and maintenance therapy [25]. Smoking cessation eventually facilitates the activity of combined steroid and β-agonist, which is hindered by smoke [26].

## 6. Conclusions

Smoking cessation treatment confirms its central role in slowing the decline in respiratory function and indeed leads to clinical-functional improvement in the short term. Treatment with triple therapy in COPD becomes more effective in the patient who achieves smoking cessation.

## Figures and Tables

**Table 1 jcm-12-00234-t001:** Baseline Data.

	M	SD
Gender (male/female)	65.0%	
Age, yr	55.5	1.6
BMI, kg/m^2^	30.7	3.5
Pack-year	25.0	2.5
Exhaled CO ppm	23.0	2.5
FEV1 l	1.12	0.2
FVC%	70.0%	3.5
FTND	5.0	1.0

FEV1: forced expiratory one second; FVC: forced vital capacity; FTND: Fagestrom Nicotine Dependence Test.

**Table 2 jcm-12-00234-t002:** Functional baseline profile for patient quitters compared with sustainers.

	Quitters	Sustainers	*p*
Age	54.5 (45.5–64)	57.5 (47–68.1)	<0.10
BMI	25.0 (21–31.2)	27.0 (23.0–32)	<0.15
M/F%	55/45%	58/42%	<0.15
Exhaled CO ppm	23.0 (15.0–25.0)	22.5 (12.0–20.0)	<0.05
eCO post	8.0 (4.0–12.0) °°	20.0 (12.0–22.0)	<0.01
FEV1 l	1.12 (1.10–1.20)	1.15 (1.10–1.30)	<0.10
FEV1 post	1.21 (1.9–2.5) ”	1.16 (1.10–1.30)	<0.01
FEV1%	48.5 (44.6–51.5)	50.2 (47.4–53.0)	<0.21
FEV1% post	55.0 (51.5–58.4) ”	51.5 (48.3–56.2)	<0.06
FVC l	1.93 (1.8–2.5)	1.95 (1.8–2.1)	<0.08
FVC post	2.00 (1.9–2.2) ”	1.91 (1.9–2.4)	<0.04
FVC%	72.1 (64.1–78.0)	73.3 (69.1–77.3)	<0.12
FVC% post	78.5 (71.3–84.2) ”	73.1 (69.5–76.2)	<0.05
CAT	12.0 (10.5–13.5)	11.5 (8.5–13.0)	<0.12
CAT post	8.5 (7.0–10.5)	11.0 (9.0–12.0)	<0.04
Walking test m	310 (300–340)	300 (280–350)	<0.10
WT post	400 (370–440) °°	315 (290–340)	<0.01
FEF 25/75 l	0.5 (0.3–0.60)	0.5 (0.4–0.7)	<0.15
FEF 25/75	0.6 (0.4–0.7)	0.4 (0.3–0.6)	<0.20

Median and Interquartile range, Mann–Whitney test for comparison between groups; Wilcoxon sign test for comparison inside the groups; **°°**
*p*<0.01; ” *p* < 0.05; CAT: COPD assessment test.

**Table 3 jcm-12-00234-t003:** Changes from baseline by smoking status between group 1 and group 2.

	Quitters 95CI	Sustainers	95CI	*p*
FEV1 l	0.09 (0.06–0.15)	0.01	(0–0.04)	<0.005
eco ppm	15.0 (13.0–16.0)	2.5	(1.5–3.5)	<0.003
WT m	90.0 (80.0–100.0)	15.0	(10.0–20.0)	<0.001

Mann–Whitney test; comparison of variations; WT: 6 min walking test.

## Data Availability

Data are unavailable due to ethical restriction.

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
