# Peer review of "Functional Benefit of Smoking Cessation and Triple Inhaler in Combustible Cigarette Smokers with Severe COPD: A Retrospective Study"

_jcm, 2022, doi:10.3390/jcm12010234_

Round 1

Reviewer 1 Report

The manuscript by Pezzuto et al describes a study which assessed the effects of smoking cessation in a group of COPD severe patients, treated with triple therapy.

The manuscript deals with a topic of relevant clinical interest, however the authors should reconsider some points. 

Major points

First of all, the authors should provide further details about study protocol and patients. Notably, they should inform about the posology (in µg) of the triple combination BDP/FF/G and about the device (MDI or MPI). Furthermore, they should provide the anthropometric characteristics (age, gender and BMI) of the two subgroups of patients (quitters vs sustainers) in table 2. Finally, they should report the FEV1 and FVC values not only in terms of absolute values, but also of percent of predicted values. In the discussion section, the authors also should discuss the limitations of the study (retrospective study, reduced number of patients, etc).       

Minor points

The authors should further clarify what means the test “a µto 4” (line 92, page 3). In table 2, there is erroneously a value of SD (2?) which refers to the first variable expressed as a percent. In table 3, the units of measure of the variables are missing. The mean forced expiratory flow between 25% and 75% of the FVC is reported as acronym MMEF in table 2 and as FEF 25/75 in the text (page 3 line 94 end line 124); authors should use a single acronym.

Lastly, a linguistic review by a native-speaking reviewer is strongly recommended.

Author Response

Dear reviewer,

thank you for your comments and suggestions.

The posology and device of the triple combination have been added.

The anthropometric characteristics of the subgroups were provided in table 2.

FVC and FEV1 as a percentage of the predicted value have been added.

The limitations of the study have been discussed.

The line 92 page 3 has been edited. Table 2 has been edited accordingly.

In table 3 the units were included . MMEF has been replaced with FEF 25/75.

A linguistic review by a native english speaker has been performed.

Reviewer 2 Report

Method- please provide a little more description on the 'several test' done and what was the counseling content

Conclusion- It must be in line with the aim of study-comparison of inhaler treatment and between habit of quitting.

Author Response

Dear reviewer,

thank you for your suggestions and comments.

The tests were better described including a description of counselling.

The conclusions have been edited accordingly.

You can find the attached revised version.

Round 2

Reviewer 1 Report

My criticisms have been adequately considered by the authors, who have appropriately revised the manuscript.